# Identification of the Components in a *Vaccinium oldhamii* Extract Showing Inhibitory Activity against Influenza Virus Adsorption

**DOI:** 10.3390/foods8050172

**Published:** 2019-05-20

**Authors:** Haruhito Sekizawa, Kazufumi Ikuta, Mayumi Ohnishi-Kameyama, Kyoko Nishiyama, Tatsuo Suzutani

**Affiliations:** 1Product Quality and Processing Division, Fukushima Agricultural Technology Centre, 116 Takakura, Hiwada, Koriyama, Fukushima 963-8041, Japan; sekizawa_haruhito_01@pref.fukushima.lg.jp; 2Department of Microbiology, Fukushima Medical University School of Medicine, 1 Hikarigaoka, Fukushima 960-1295, Japan; ikutak@tohoku-mpu.ac.jp (K.I.); kyoko@fmu.ac.jp (K.N.); 3Division of Microbiology, Tohoku Medical and Pharmaceutical University, Fukumuro 1-15-1, Miyagi, Japan; 4Food Research Institute, NARO, Kannondai 2-1-12, Tsukuba, Ibaraki, Japan; kameyama@affrc.go.jp

**Keywords:** anti-influenza virus, blueberry, functional foods, polyphenol

## Abstract

We previously reported that extracts from plants of the Ericaceae genus *Vaccinium*, commonly known as the kind of blueberry, inhibited the early steps of influenza virus (IFV) infection to host cells, and that the activity was correlated with the total polyphenol content. Particularly potent inhibitory activity was observed for *Vaccinium oldhamii*. In this study, we identified the active components in *Vaccinium oldhamii* involved in the inhibition of IFV infection. We sequentially fractionated the *Vaccinium oldhamii* extract using a synthetic adsorbent resin column. High inhibitory activity was observed for the fractions eluted with 30%, 40%, and 50% ethanol, and three peaks (peak A, B, and C) considered to represent polyphenols were identified in the fractions by HPLC analysis. Among these peaks, high inhibitory activity was detected for peak A and B, but not for peak C. These peaks were analyzed by LC/MS, which revealed that peak A contained procyanidin B2 and ferulic acid derivatives, whereas peak B contained two ferulic acid *O*-hexosides, and peak C contained quercetin-3-*O*-rhamnoside and quercetin-*O*-pentoside-*O*-rhamnoside. It is already known that these polyphenols have anti-IFV activity, but we speculate that ferulic acid derivatives are the major contributors to the inhibition of the early steps of IFV replication, such as either adsorption or entry, observed for *Vaccinium oldhamii*.

## 1. Introduction

The blueberry is popular around the world as a functional fruit. It is reported to promote the recovery of sight in the dark [1] and possess protective effects in retinal pigment and corneal epithelial cells [2,3]. Blueberries contain a large amount of various polyphenols, such as anthocyanin, which are thought to be a source of antioxidants for the removal of reactive oxygen species in the body [4,5], and are reported to suppress oxidative stress and cirrhosis of the liver [6]. However, little is known about the antiviral activities of blueberries. 

We previously reported the antimicrobial effects of blackcurrant (*Ribes nigrum*). The blackcurrant extract demonstrated antiviral activities against herpes simplex virus (HSV) type 1 [7], respiratory syncytial virus [8], and influenza virus (IFV) types A and B [9,10], as well as antibacterial activities against *Streptococcus pneumoniae* and *Haemophilus influenzae* type b [8]. In order to clarify the anti-IFV effect of blueberries, we examined the inhibitory activity against IFV infection of 36 kinds of fruit belonging to the Ericaceae genus *Vaccinium* [11]. As a result, two kinds of northern highbush blueberries, seven kinds of rabbit-eye blueberries, and a wild blueberry *Vaccinium oldhamii* were found to possess anti-IFV activity. The inhibitory effects against IFV infection were positively correlated with the total polyphenol content, with the highest inhibitory effect confirmed in *Vaccinium oldhamii*. Here, we further pursue our studies by fractionating the polyphenols from a *Vaccinium oldhamii* extract that was found to show an inhibitory effect against IFV infection.

## 2. Materials and Methods 

### 2.1. Specimens

*Vaccinium oldhamii* is a wild blueberry found in Japan, China and the southern areas of the Korean Peninsula. It is known as ‘Natsuhaze’ in Japan. We harvested the fruit used in this study at Fukushima, Japan in November 2010.

### 2.2. Separation of Vaccinium oldhamii by Synthetic Absorbent

The *Vaccinium oldhamii* fruit was frozen, dried, homogenized, and immersed in 80% ethanol. After letting the sample stand overnight at 5 °C, the supernatant was obtained as an ethanol extract by centrifugation at 8000× *g* for 10 min and concentration under reduced pressure. 

The concentrated ethanol extract (equivalent to 5 g of dried fruit) was then diluted with 50 mL of distilled water, and applied to column chromatography using a synthetic absorbent resin (DIAION HP-20, 20 g, Mitsubishi Chemical Co., Ltd., Tokyo, Japan) packed in a glass column (20 mm in diameter, 300 mm in length). The eluate obtained using distilled water or various concentrations of ethanol (10%, 20%, 30%, 40%, 50% or 80%) was fractionated, concentrated, and weighed.

Each fraction was dissolved in 20% dimethyl sulfoxide (DMSO, Nacalai Tesque, Inc., Kyoto, Japan) to adjust the concentration to 1 mg/mL, and then tested for total polyphenol and anthocyanin content, as well as for inhibitory activity against IFV infection.

### 2.3. Separation of Vaccinium oldhamii by HPLC

The *Vaccinium oldhamii* fruit was frozen, thawed, homogenized, added to 0.1% pectinase (pectinase SS, Yakult Pharmaceutical Industry Co., Ltd., Tokyo) and subsequently incubated at 40 °C for 2 h. The supernatant obtained by centrifugation at 8000× *g* for 10 min was adsorbed to a synthetic absorbent, washed with distilled water to remove water-soluble contaminants, and the adsorbed materials were then eluted with 80% ethanol. The eluate was lyophilized by removal of ethanol under reduced pressure, and the resultant extract was dissolved in distilled water to a final concentration of 10 mg/mL, and then the peak of polyphenol, which was presumed to have high IFV infection inhibitory activity contained in the extract, was fractionated using high-performance liquid chromatography (HPLC). The obtained fractions were weighed after concentration to dryness under reduced pressure, and then dissolved in 20% DMSO (Nacalai Tesque) to adjust the concentration to 1 mg/mL. They were then tested for total polyphenol content and inhibitory activity against IFV infection.

### 2.4. Measurement of Total Polyphenol Content in Vaccinium oldhamii

Total polyphenol content was measured using the Folin-Ciocalteu method [12]. Briefly, 60 μL of distilled water, 10 μL of sample and 15 μL of twice-diluted Folin-Ciocalteu reagent were mixed, stirred in 96-well microplates, and incubated at room temperature for 5 min. The solutions were then added to 75 μL of 2% sodium carbonate solution and further incubated at room temperature for 15 min. Absorbance at 750 nm was then measured and the total polyphenol content was given as a gallic acid (Wako Pure Chemical Industries, Ltd., Osaka, Japan) equivalent.

### 2.5. Measurement of Total Anthocyanin Content in Vaccinium oldhamii

Total anthocyanin content in *Vaccinium oldhamii* was measured using the absorbance method [13]. Briefly, 120 μL of 5% trifluoroacetic acid and 30 μL of sample were mixed and stirred in 96-well microplates. Absorbance at 520 nm was then measured and the total anthocyanin content was given as a cyanidin-3-glucoside chloride (Wako Pure Chemical Industries, Ltd., Osaka, Japan) equivalent.

### 2.6. Cells and Viruses

Madin-Darby canine kidney (MDCK) cells (Rational Drug Design Laboratories., Fukushima, Japan) were used for the measurement of the antiviral effects of the samples. The MDCK cells were cultured in Dulbecco’s modified Eagle’s medium (DMEM, Nissui Pharmaceutical Co., Ltd., Tokyo, Japan) containing 4% fetal calf serum and antibiotics (300 μg/mL of streptomycin, 300 U/mL of penicillin and 1 μg/mL of amphotericin B). 

The influenza virus (A/Yamagata/165/2009(H1N1pdm09)), which was isolated in Yamagata, Japan in the 2009–2010 season, was provided by the Yamagata Prefectural Institute of Public Health. The IFV-infected MDCK cell supernatant was harvested as a virus suspension and stored at −80 °C until use.

### 2.7. IFV Adsorption Inhibition Assay

IFV infection inhibitory activities were measured according to previous reports [11,14]. The MDCK cells were cultured in 12-well plates until confluence. IFV was diluted with DMEM at 10,000, 1000, or 100 pfu per well with the *Vaccinium oldhamii* fraction at 0 (as a control), 25, 75 or 250 μg/mL in DMEM that also contained 4% bovine serum albumin (DMEM/BSA). The virus was adsorbed to the MDCK cells at room temperature for 5 min with stirring. The infection time of 5 min was set in consideration of the application of *Vaccinium oldhamii* to certain foods such as lozenges and chewing gum. The virus/specimen mixture was removed, and the culture was washed and overlaid with DMEM/BSA containing 1% Ultrapure Agarose (Thermo Fisher Scientific, Waltham, MA, USA) with 1 μg/mL of TPCK-trypsin (Sigma-Aldrich, Co., St. Louis, MO, USA). After incubation for 4 days at 37 °C in a CO_2_ incubator, cells were fixed with 10% formaldehyde in phosphate-buffered saline and stained with 0.1% crystal violet. The inhibitory activity against IFV infection was calculated from the number of plaques reduced by the supernatant from *Vaccinium oldhamii*, and expressed as a 50% inhibitory concentration (IC_50_). Results were expressed as the average of three independent experiments.

### 2.8. Cell Toxicity Assay

The effects of the test fractions on the replication of MDCK cells were evaluated as previously described [15]. The MDCK cells were seeded in 48-well plastic plates at 5 × 10^3^ per well. The cells were refed one day later with DMEM plus 4% fetal calf serum containing an appropriate amount of the test fractions. After incubation for 2 days, cells were dispersed by treatment with trypsin, and the number of viable cells was counted. The 50% cytotoxic concentration (CC_50_) was determined graphically and CC_50_ and IC_50_ ratios were also determined as the selectivity index (SI).

### 2.9. ESI–LC/MS Analysis

Electrospray ionization–liquid chromatography/mass spectrometry (ESI–LC/MS) was performed using an LTQ Orbitrap Veros Pro mass spectrometer (Thermo Fisher Scientific) equipped with an Ultimate 3000 RSLC (Thermo Fisher Scientific). Fractions obtained by HPLC purification were injected into a C18 column (ACQUITY UPLC HSS C18, 1.8 μm, 2.1 mm × 100 mm, Waters, Milford, MA, USA) and separated using solution A (0.1% formic acid) and B (acetonitrile containing 0.1% formic acid) at 40 °C and a flow rate of 0.3 mL per minute. The concentration of B was maintained at 5% from 0 to 1 min, gradually increased to 80% over the subsequent 9 min, and then decreased to 50% over the next 10 min. Thereafter, the column was equilibrated at the initial conditions (5% B) for a further 10 min. The eluates passing through the Photodiode Arraydetector were ionized and introduced into the mass spectrometer. The parameters were set as follows: spray voltage 3.0 kV, source heater temperature 400 °C, capillary temperature 230 °C, sheath gas flow rate 40 (arbitrary units), auxiliary gas flow rate 5 (arbitrary units), resolution 10,000, and mass range *m*/*z* 100–1000.

## 3. Results

### 3.1. Relationships among the IFV Adsorption Inhibitory Activities and Polyphenols in the Vaccinium oldhamii Fraction

The inhibitory activity against IFV infection of *Vaccinium oldhamii* fractionated by synthetic absorbent resin was measured as displayed in Table 1. The water fraction (fraction #0) gave the highest yield among all fractions and was supposed to contain water-soluble components, such as saccharides, which cannot interact with the synthetic adsorbents. *Vaccinium oldhamii* fruit contains large amounts of anthocyanins and chlorogenic acid. Many anthocyanins were eluted with 10% and 20% ethanol by column chromatography using synthetic absorbent resin. No inhibitory activity against IFV infection was observed in the highly hydrophilic fractions #0 and #1, although fraction #1 showed intense peaks in the early phase of the HPLC chromatogram, shown in Figure 1, lower panel.

High IFV infection inhibitory activities were observed for fractions #3, #4, and #5 (the 30%, 40%, and 50% ethanol fractions, respectively) with intense peaks detected at 35–45 min on the HPLC chromatograms for these fractions, as seen in Figure 1. Polyphenols were eluted in fractions #1 to #5. However, anthocyanin was eluted in only limited yields in fractions #1 and #2. Therefore, the major contributors to the IFV infection inhibitory activity of *Vaccinium oldhamii* were considered to be polyphenols other than anthocyanins, and the strength of the antiviral activity is speculated to vary with polyphenol type. A high selectivity index value was observed for fractions #3 and #4.

In order to identify the components contained in these fractions that inhibit IFV infection, the peaks were separated by HPLC, and fractions A, B, and C obtained, as shown in Figure 2. The yields of fraction A, B and C, as listed in Table 2, were 7.6, 3.1, and 4.2 mg, respectively. Although fraction C contained the highest amount of total polyphenol, its IFV infection inhibitory activity was the lowest among the three fractions, being approximately 1/6 that of both fractions A and B. The polyphenols in fractions A and B were therefore considered to be candidates for the main active components in *Vaccinium oldhamii* responsible for its inhibition of IFV infection.

*Vaccinium oldhamii* contains large amounts of anthocyanins and chlorogenic acid in addition to hydrophilic compounds eluted early during the column chromatography by synthetic adsorbents. Polyphenol peaks with high intensities were observed at 35–45 min.

### 3.2. Mass Spectrometry of Vaccinium oldhamii HPLC Fractions

Fractions A, B, and C of *Vaccinium oldhamii*, obtained by HPLC as described in the previous section, were analyzed by ESI–LC/MS. The profiles of the chromatograms differed from those obtained using conventional HPLC as different LC columns were used, and the chromatograms were prepared by the integration of UV/Vis absorbance in the range from 220 to 750 nm. That is, more peaks were detected in the UV/Vis chromatograms obtained by ESI–LC/MS. The major components in fractions A, B, and C gave peaks #1, #2, and #3; peaks #4 and #5; and peaks #6 and #7, respectively, as shown in Figure 3.

The chromatograms were integrated for absorbance in the range from 220 to 750 nm. Fraction A contained three peaks, and fractions B and C possessed two peaks each.

Peak #1 at 7.10 min showed as an intense peak at *m*/*z* 577 and 579 in the negative-ion and positive-ion mode, respectively, as listed in Table 3. The MS/MS (Tandem mass spectrometry) experiment under [M − H]^−^ gave product ions at *m*/*z* 451, 425, 407, and 289, and under [M + H]^+^ gave product ions at *m*/*z* 427, 429, and 291. Further, the UV/Vis spectrum of peak #1 showed a maximum absorption at 279 nm. These results indicated that peak #1 was an epicatechin dimer, such as procyanidin B1 or B2. We next compared the mass and MS/MS spectra as well as the UV/Vis spectra of a commercially available procyanidin B2 standard and peak #1, and confirmed peak #1 to be procyanidin B2.

Peak #2 at 7.45 min and peak #3 at 7.99 min showed almost the same mass and MS/MS spectra in the negative-ion mode. These peaks gave [M − H]^−^ at *m*/*z* 355, and product ions at *m*/*z* 295 ([M − H − 60]^−^), 235 ([M − H − 120]^−^), 217 ([M − H − 120 − 18]^−^), 193 ([M − H − 162]^−^), and 175 ([M − H − 162 − 18]^−^). The UV/Vis spectra were also almost the same and gave a maximum absorption at around 329 nm. The positive-ion mass and MS/MS spectra did not give informative peaks. It was reported that ferulic acid derivatives in black carrot showed [M − H]^−^ at *m*/*z* 355, product ions at *m*/*z* 295, 217, 193, 175, and 134, and λ max at 330 nm [16]. Therefore, these peaks were speculated to be ferulic acid derivatives.

Peaks #4 (11.05 min) and #5 (11.46 min) of fraction B also showed [M − H]^−^ at *m*/*z* 355. However, the MS/MS spectra were different from those of peaks #2 and #3. [M − H]^−^ at *m*/*z* 355 gave an intense product ion at *m*/*z* 193 ([M − H − 162]^−^), with no other intense peaks observed. This pattern was previously reported for ferulic acid hexoside in dried plums by Fang [17] and in herbs by Vallverdú-Queralt [18]. These data indicated that the major peaks in fraction B were ferulic acid O-hexosides.

In the negative-ion mass spectra of peaks #6 (12.47 min) and #7 (12.55 min), intense peaks were observed at *m*/*z* 447 and *m*/*z* 579, respectively. In the positive-ion mode, peaks #6 and #7 gave [M + H]^+^ at *m*/*z* 449 and *m*/*z* 581, respectively, and the dominant product ions were observed at *m*/*z* 303 ([M + H − 146]^+^) and *m*/*z* 449 ([M + H − 146]^+^), respectively. As for the MS/MS spectrum of peak #7, a similar fragmentation pattern to that of peak #6 was observed. That is, product ions other than the one at *m*/*z* 449 were observed at *m*/*z* 431 ([M + H − 132 − 18]^+^), 413 ([M + H − 132 – 18 × 2]^+^), 345 and 303 ([M + H − 132 − 146]^+^). Further, the UV/Vis spectra of peaks #6 and #7 showed maximum absorption around 349 nm. These data suggested that peaks #6 and #7 were quercetin derivatives, such as quercetin-3-*O*-rhamnoside and quercetin-*O*-pentoside-*O*-rhamnoside, respectively [19].

## 4. Discussion and Conclusions

Berry fruits such as blackcurrant [8,9,10], aronia [20], elderberry [21], and cranberry [22] have been reported to have anti-IFV activity, and the polyphenols contained in these fruits are believed to contribute to these antiviral activities. Applications for the prevention of influenza infection using these fruits are expected in the future. The purpose of our research is to aid in the development of foods, such as troches and chewing gum, which can prevent IFV infection at the mucosal surface. In order to evaluate the inhibitory effect on the early steps in IFV infection, we developed an assay in which the sample and IFV were applied to an MDCK cell monolayer simultaneously, washed out after a 5-min adsorption period and incubated with agar medium prior to the counting of plaque numbers [8]. This assay revealed that two fractions, A and B, isolated from *Vaccinium oldhamii*, showed potent inhibitory activity against IFV infection, with procyanidin B2 and ferulic acid derivatives identified as the major active components in these fractions. Moreover, quercetin derivatives were identified from the fraction showing weak anti-IFV activity.

Ferulic acid and its derivatives have been reported to bind to neuraminidase and inhibit the initial stage of IFV infection, with its derivatives in particular showing a more potent effect [23]. The two fractions A and B obtained in this study contained ferulic acid derivatives. Therefore, these results suggested that ferulic acid derivatives contribute to the IFV infection inhibitory activity of *Vaccinium oldhamii*.

Procyanidin B2 was previously reported to inhibit IFV replication in cells [24], but no effect was observed at the adsorption step of IFV infection [25]. Therefore, the high inhibitory activity against IFV infection of fraction A observed in this study is speculated to be due to the ferulic acid derivatives rather than procyanidin B2.

The inhibitory activity of fraction C was weak, with the major component in the fraction found to be quercetin glycoside. The anti-IFV activity of quercetin rhamnoside was reported previously, but this activity was due to the suppression of IFV replication in cells, not by the inhibition of virus infection to host cells [26]. In order to suppress IFV replication, quercetin rhamnoside is required to act continuously in IFV-infected cells.

From these results, some polyphenols having anti-IFV activity were found to be contained in *Vaccinium oldhamii*, but it was clarified that the major contributors to the inhibition of the early stage of IFV infection were ferulic acid derivatives. However, we should also consider the possibility of a synergistic effect among various components. For example, the anti-IFV activity of catechin was enhanced when multiple catechins were mixed [27]. Therefore, the strong inhibitory effect of *Vaccinium oldhamii* on IFV infection might not be due only to the effect of the ferulic acid derivatives but also a synergistic effect among the various polyphenols contained in the fruit. Further study is required to clarify the network of components in the plant that might act synergistically. Also, in recent years, research has been actively conducted on processing methods for suppressing the decrease in active components, such as polyphenols [28,29], so it is thought that the opportunities for utilizing our research results will increase in the future. The goal of this study is to prevent the spread of influenza infection through the utilization of everyday foods, such as fruits, rather than by medicines. We hope that this research will make a significant contribution to that purpose.

## Figures and Tables

**Figure 1 foods-08-00172-f001:**
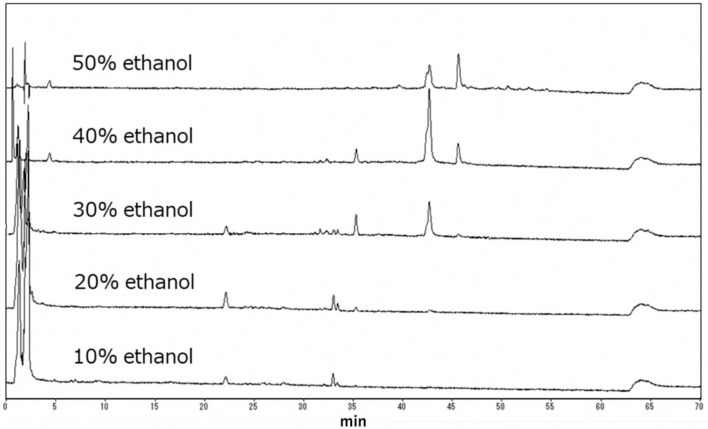
High-performance liquid chromatography (HPLC) chromatograms for the *Vaccinium oldhamii* fractions separated by synthetic adsorbents.

**Figure 2 foods-08-00172-f002:**
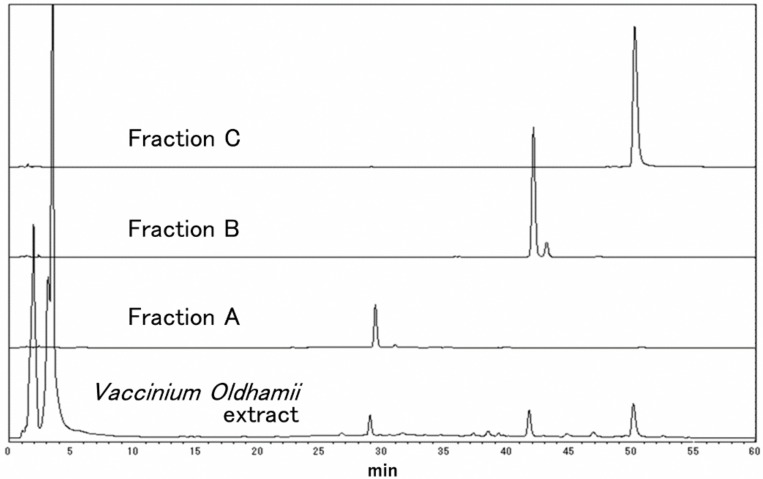
HPLC chromatograms for the *Vaccinium oldhamii* fractions separated by C18 column. Fractions A, B, and C were obtained from the fractions eluted with 30%, 40%, and 50% ethanol in Figure 1. Anthocyanin and chlorogenic acid were eluted in the earlier fractions, and they are not included in the three fractions (A to C).

**Figure 3 foods-08-00172-f003:**
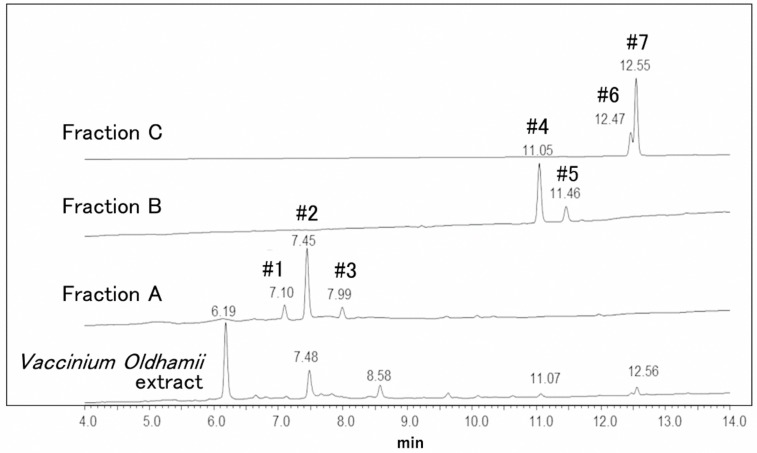
HPLC chromatograms for the fractions separated by electrospray ionization–liquid chromatography/mass spectrometry (ESI–LC/MS).

**Table 1 foods-08-00172-t001:** Separation of *Vaccinium oldhamii* by synthetic absorbents.

Fraction No.	Elute Solvent	Volume of Solvent	Weight of Fraction	Total Polyphenol Content	Total Anthocyanin Content	50% Adsorption Inhibitory Concentration (IC_50_)	50% Cytotoxic Concentration (CC_50_)	Selectivity Index
		(mL)	(mg)	(µg/mL)	(µg/mL)	(µg/mL)	(µg/mL)	(CC_50_/IC_50_)
	Fruit (dry)	-	5000	-	-	-	-	-
	Extract	-	3077	423	297	72	159	2.2
0	Water	200	2867	28	2	ND	>400	ND
1	10% ethanol	100	76	415	439	ND	236	ND
2	20% ethanol	100	114	445	377	159	>400	>2.5
3	30% ethanol	100	47	400	149	38	251	6.6
4	40% ethanol	100	20	327	80	22	160	7.3
5	50% ethanol	100	3	330	90	65	78	1.2
6	80% ethanol	200	4	191	91	85	140	1.6

The initial material was extracted from 5 g of dried fruit using 80% ethanol, and its amount was calculated based on the amount of insoluble material. The extract and each fraction were adjusted to 1 mg/mL. ND—not detected.

**Table 2 foods-08-00172-t002:** Separation of *Vaccinium oldhamii* by HPLC. IFV—influenza virus.

Fraction	Fraction Yield	Total Polyphenol Content	IFV Adsorption Inhibitory Activity (IC_50_)
	(mg)	(µg/mL)	(µg/mL)
A	7.6	228	38
B	3.1	278	40
C	4.2	371	238

Fraction A, B, and C were included in the 30% to 50% ethanol fractions listed in Table 1, which showed high IFV adsorption inhibitory activity.

**Table 3 foods-08-00172-t003:** Observed peaks in ESI–LC/MS analysis.

Fraction	Peak	Retention Time	UV (*λ*max)	ESI Mode	Precursor Ion (Relative Intensity, %)	Product Ion	Putative Compound
	No.	(min)	(nm)		(*m*/*z*)	(*m*/*z*)	
A	#1	7.10	279	(−)	451(32), **425**(100), 407(41), 289(16)	451, 425, 407, 289	procyanidin B2
				(+)	**427**(100), 409(63), 291(33)	427, 429, 291	
	#2	7.45	329	(−)	295(8), 235(12), 217(55), **193**(100), 175(47)	295, 235, 217, 193, 175	ferulic acid derivatives
				(+)	no information	no information	
	#3	7.99	325	(−)	295(7), 235(12), 217(55), **193**(100), 175(45)	295, 235, 217, 193, 175	ferulic acid derivatives
				(+)	no information	no information	
B	#4	11.05	315	(−)	**193**(100)	193	ferulic acid *O*-hexosides
				(+)	**195**(100), 163(45)	195, 163	
	#5	11.46	307	(−)	**193**(100)	193	ferulic acid *O*-hexosides
				(+)	**195**(100), 163(50)	195, 163	
C	#6	12.47	349	(−)	**301**(100)	301	quercetin *O*-rhamnoside
				(+)	**303**(100)	303	
	#7	12.55	349	(−)	**300**(100)	300	Quercetin *O*-pentoside *O*-rhamnoside
				(+)	**449**(100)	449	

Analysis was performed in the negative- and the positive-ion modes. Fraction A included procyanidin B2 and ferulic acid derivatives, and fractions B and C included ferulic acid hexosides and quercetin glycoside, respectively.

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
