# Peer review of "Identification of the Components in a Vaccinium oldhamii Extract Showing Inhibitory Activity against Influenza Virus Adsorption"

_foods, 2019, doi:10.3390/foods8050172_

Round 1

Reviewer 1 Report

The manuscript, entitled “Identification of the components in a Vaccinium oldhamii extract showing inhibitory activity against influenza virus adsorption” presented interesting data. Basically, this manuscript contains some interesting results and scientific merit. From this point of view, the investigation of this paper should be encouraged. The work is original as far as I can ascertain. The subject is relevant to the field of the Foods. Although the study is not novel, the finding is interesting and the observations do support the conclusion. I recommend the paper is publishable

Author Response

Reviewer#1

The manuscript, entitled “Identification of the components in a Vaccinium oldhamii extract showing inhibitory activity against influenza virus adsorption” presented interesting data. Basically, this manuscript contains some interesting results and scientific merit. From this point of view, the investigation of this paper should be encouraged. The work is original as far as I can ascertain. The subject is relevant to the field of the Foods. Although the study is not novel, the finding is interesting and the observations do support the conclusion. I recommend the paper is publishable

> Thank you very much for reviewing this paper and recognizing the significance of our research. We would like to apply these results to future food development.

Reviewer 2 Report

Introduction

The article deals with the identification of components of the fruits of Vaccinium oldhamii, to which could be attributed the observed inhibitory activity against the infection (early phase) by the influenza virus.

This study shows clear continuity with previous studies (and articles) carried out by the same research group, and is sufficiently original in the search for specific bioactive compounds to which the considered effect could be attributed.

The main result - in brief, attribution to ferulic acid derivatives more than to anthocyanins and quercetin (and procyanidin), despite leaving open the topic of possible synergism, needing further investigation - is quite remarkable, also because it could be extended to other plant products. 

The manuscript complies with the scope and aims of this Journal, as it foresees the preparation of food products with antiviral properties, as well as the experimental design fits with this goal, for example with the 5-minutes adsorption period.

The language and the style are substantially fine and attractive, despite few marginal errors (see specific comments below).

General comments

Two general comments, aimed at completeness:

1) Are the obstained results in principle extendable to other food products? That is, to other products with substantial concentration of ferulic acid derivates? If true, this should be emphasized.

2) Well for troches and chewing gum (line 241), but what about, for example, jiuces and purees? If the Authors think that also juices and purees, made of the same fruit, could be useful for prevention of the viral infection, then some reference could be done to recent important progresses on the solvent-free extraction and preservation of polyphenols in purees from blueberries and cranberries, such as by means of processes based on controlled hydrodynamic cavitation (doi:10.1016/j.jfoodeng.2014.11.016 and doi:10.1016/j.lwt.2016.10.060).

Specific comments

Line 56. "2.2. Separation of Vaccinium oldhamii by synthetic absorbent". Is there a reference available for this method?

Line 66. "1 mg/mL". Sure about the unit?

Line 68. "2.3. Separation of Vaccinium oldhamii by HPLC". Although common enough, is there a reference available for this method?

Lne 69. "added to 0.1% pectinase". Explain the reason for use of pectinase enzyme. Is it necessary in order to speed up the extraction of polyphenols and anthocyanins?

Line 75. "10 mg/mL". Sue about the unit?

Line 119. "2.8. Cell toxicity assay". Is there a reference available for this method?

Line 125. "2.9. ESI-LC/MS Analysiss". Change to "2.9. ESI-LC/MS Analysis". Moreover, is there a reference available for this method?

Line 157. "fraction". Change to "fractions".

Table 3, second row, last column. "acid derivatives". Likely, it is "ferulic acid derivatives"?

Line 236. "4. Discussion". Change to "4. Discussion and conclusions".

Line 245. "numbers8.". Did you want to make reference to [8]?

Line 256. "faction". Change to "fraction".

Line 267. "[26]" and respective reference at lines 344-345. Uncorrect reference. It should be: (Yang, Z. F.; Bai, L. P.; Huang, W. B.; Li, X. Z.; Zhao, S. S.; Zhong, N. S.; Jiang, Z. H. Comparison of in vitro antiviral activity of tea polyphenols against influenza A and B viruses and structure-activity relationship analysis. Fitoterapia 2014, 93, 47–53, doi:10.1016/j.fitote.2013.12.011).

Line 269. "Further the study". Change to "Further study".

Author Response

Reviewer#2

1)  Are the obstained results in principle extendable to other food products? That is, to other products with substantial concentration of ferulic acid derivates? If true, this should be emphasized.

> We observed the same anti-influenza virus activities in blueberry fruits. However, the two ferulic acid derivatives have not been detected in blueberries. This indicated that different fruits contain different anti-influenza virus factors. Moreover, the anti-influenza activities of the purified components were much weaker than that of the crude juice. This result indicated that various factors act synergistically as described in the “Discussion and Conclusions” section (line 264-271). Therefore, we could not emphasize the extension of our results.

2) Well for troches and chewing gum (line 241), but what about, for example, iuces and purees? If the Authors think that also juices and purees, made of the same fruit, could be useful for prevention of the viral infection, then some reference could be done to recent important progresses on the solvent-free extraction and preservation of polyphenols in purees from blueberries and cranberries, such as by means of processes based on controlled hydrodynamic cavitation (doi:10.1016/j.jfoodeng.2014.11.016 and doi:10.1016/j.lwt.2016.10.060).

> As we also consider the application of the fruit to various food products other than troches and chewing gum, we added a description in the last part (line 271-274) and cited the papers you introduced as references [27, 28]. Thank you very much for your important discussion.

Specific comments

Line 56. "2.2. Separation of Vaccinium oldhamii by synthetic absorbent". Is there a reference available for this method?

> It is a general method described in the resin manufacturer's manual, so we didn’t cited any papers.

Line 66. "1 mg/mL". Sure about the unit?

> Yes. Adjusted as shown.

Line 68. "2.3. Separation of Vaccinium oldhamii by HPLC". Although common enough, is there a reference available for this method?

> We determined the detailed conditions using a widely used method. The point was that we used ethanol as a solvent as it is safe to eat. Separation conditions were described in the manuscript, so we didn’t cite any paper.

Line 69. "added to 0.1% pectinase". Explain the reason for use of pectinase enzyme. Is it necessary in order to speed up the extraction of polyphenols and anthocyanins?

>We used pectinase to obtain a large volume of juice. The recovery rate of polyphenols is increased by the decomposition of pectin contained in the fruit with pectinase. The viscosity of the juice was decreased by this treatment as well.

Line 75. "10 mg/mL". Sue about the unit?

>Yes. It is a high concentration, but was adjusted as shown.

Line 119. "2.8. Cell toxicity assay". Is there a reference available for this method?

>We cited a paper as a reference for this method and added a sentence (line 120-121, ref. 15).

Line 125. "2.9. ESI-LC/MS Analysiss". Change to "2.9. ESI-LC/MS Analysis". Moreover, is there a reference available for this method?

> Our method was slightly modified from the usual method. For the separation of the fruit extract by HPLC, acetonitrile is generally used as a solvent. However, we used ethanol[GH1] , so there is no reference. However, the detailed conditions were considered in advance

Line 157. "fraction". Change to "fractions".

> We corrected this as suggested. Thank you very much for pointing out our mistake.

Table 3, second row, last column. "acid derivatives". Likely, it is "ferulic acid derivatives"?

> We corrected it according to your suggestion. Thank you very much.

Line 236. "4. Discussion". Change to "4. Discussion and conclusions".

> We corrected this as suggested. Thank you very much.

Line 245. "numbers8.". Did you want to make reference to [8]?

> We corrected this according to your suggestion. Thank you very much for pointing out our mistake.

Line 256. "faction". Change to "fraction".

> We corrected this as suggested. Thank you very much.

Line 267. "[26]" and respective reference at lines 344-345. Uncorrect reference. It should be: (Yang, Z. F.; Bai, L. P.; Huang, W. B.; Li, X. Z.; Zhao, S. S.; Zhong, N. S.; Jiang, Z. H. Comparison of in vitro antiviral activity of tea polyphenols against influenza A and B viruses and structure-activity relationship analysis. Fitoterapia 2014, 93, 47–53, doi:10.1016/j.fitote.2013.12.011).

> Although “[26]” (Song JM.; Kwang HL.; Seong BL. Antiviral effect of catechins in green tea on influenza virus. Antivir Res. 2005, 68, 66-74.) is a report on tea catechins, it is reported that mixed catecins were more effective than isolated catechins. I think that the same effect may be obtained for fruits.

Also, the previous manuscript contained references that were not necessary. We have deleted these.

Line 269. "Further the study". Change to "Further study".

> We corrected this as suggested. Thank you very much.

 [GH1]OK?